# Ocular Vascular Diseases: From Retinal Immune Privilege to Inflammation

**DOI:** 10.3390/ijms241512090

**Published:** 2023-07-28

**Authors:** Xudong Wang, Tianxi Wang, Enton Lam, David Alvarez, Ye Sun

**Affiliations:** 1Department of Ophthalmology, Harvard Medical School, Boston Children’s Hospital, Boston, MA 02115, USA; wangxudong12@gmail.com (X.W.);; 2Department of Immunology, Harvard Medical School, Boston, MA 02115, USA

**Keywords:** retina, immune privilege, inflammation, retinopathy of prematurity, age-related macular degeneration, diabetic retinopathy

## Abstract

The eye is an immune privileged tissue that insulates the visual system from local and systemic immune provocation to preserve homeostatic functions of highly specialized retinal neural cells. If immune privilege is breached, immune stimuli will invade the eye and subsequently trigger acute inflammatory responses. Local resident microglia become active and release numerous immunological factors to protect the integrity of retinal neural cells. Although acute inflammatory responses are necessary to control and eradicate insults to the eye, chronic inflammation can cause retinal tissue damage and cell dysfunction, leading to ocular disease and vision loss. In this review, we summarized features of immune privilege in the retina and the key inflammatory responses, factors, and intracellular pathways activated when retinal immune privilege fails, as well as a highlight of the recent clinical and research advances in ocular immunity and ocular vascular diseases including retinopathy of prematurity, age-related macular degeneration, and diabetic retinopathy.

## 1. Introduction

The retina is a specialized tissue in the eye responsible for encoding visual information received from the outside environment. Inside the highly structured complex, retinal neurons make synaptic connections in two interjacent layers, the outer plexiform layer (OPL) and the inner plexiform layer (IPL); their cell bodies are arranged in three distinct layers, photoreceptors in the outer nuclear layer (ONL), horizontal/bipolar/amacrine/Müller cells in the inner nuclear layer (INL), and amacrine/ganglion cells in the ganglion cell layer (RGC), while astrocytes, microglial cells, and retinal vasculature (composed of endothelial cells and pericytes) are embedded throughout the retina (Figure 1). Visual function in the eye is supported by two separate vascular systems: the choroidal vessels supply the retinal pigment epithelium (RPE) and outer retina, and three layers (superficial, intermediate, and deep) of the retinal vasculature supply the inner retina (Figure 1). To protect vital structures and functionality of the eye, mammals have evolved mechanisms to cope with internal and external threats to the ocular surface while maintaining an anti-inflammatory and immunosuppressive microenvironment [1].

A successful inflammatory response is a key innate and adaptive immune response triggered by pathogens, damaged cells, and toxic compounds to eliminate the initial cause of injury, clear the necrotic cells, and repair the destructive tissue through initiating wound healing and angiogenesis [2,3,4]. However, chronic inflammation is detrimental to health [5], and eye inflammation is no exception [6,7]. Pathologies in vascular systems can cause significant visual impairment, such as retinopathy of prematurity (ROP), vascular age-related macular degeneration (AMD), and proliferative diabetic retinopathy (PDR). There is extensive evidence correlating inflammatory factors with both clinical and experimental retinopathies [8,9,10,11], of which microglia stand at the center and play an important role in retinal inflammation through their immune surveillance function. Inflammation is clearly involved in the pathogenesis of ocular vascular diseases, but the specific cellular origin of inflammation and underlying signaling pathways are not well understood. In this review, we focus on immune privilege, inflammation, and microglia in the retina and discuss the major inflammatory and pathological mechanisms in vascular eye diseases.

## 2. Immune Privilege in the Healthy Eye

The concept of immune privilege arose from early studies showing that allografts can survive for a prolonged period if transplanted into specific sites of the body, such as the brain and anterior chamber of the eye [12]. The most beneficial purpose of immune privilege is to safeguard the vulnerable and fragile cells in a specific tissue microenvironment. Immune privilege can be broken under some pathological circumstances (infection, advanced DR, or AMD), by potentially shielding microbes in the local microenvironment from additional systemic innate and adaptive immune defenses [13,14] or by breakdown or degeneration of RPE cells (the physical barrier of immune privilege). On the other hand, immune privilege against inflammatory insults may develop antiretinal autoimmunity, in which peripheral tolerance mechanisms may not operate efficiently for ocular antigens [15]. As an extension of the brain, the retina is an immune privileged site in the eye [16,17,18,19]. For the maintenance of integrity of the retina and avoidance of any inflammatory risks, the immune privilege mechanism is developed for the protection of vulnerable retinal neurons with relatively poor renewability and low repair capacity. The literature contains excellent reviews on ocular immune privilege—its history, mechanism, and its applications in corneal transplantation [20,21,22,23,24].

Since the first proposed existence of an immune privileged site in the eye—the anterior chamber in 1948 by Medawar [12]—a variety of immune cells, pathways, and molecules have been found to be involved in ocular immune privilege (Figure 2). Several mechanisms, both active and passive, contribute to immune privilege: (i) The blood–retina barrier (BRB) and blood–aqueous barrier serve as physical barriers to isolate the interior of the eye from systemic immune circulation; (ii) A variety of endothelial cells, immune cells (such as natural killer cells and dendric cells), and retinal neuronal cells provide the second layer of protection by releasing immunosuppressive factors to inhibit effector T-cell activities and inflammatory responses inside the eye; (iii) When antigens enter into the eye, antigenic proteins in the anterior chamber, vitreous cavity, and subretinal space cause anterior-chamber-associated immune deviation (ACAID), as characterized by strong immune suppression through CD4^−^ CD8^+^ regulatory T cells as well as Th1-cell- and Th2-cell-based immune-mediated inflammation, acting as a third layer of protection [15,25]. Figure 3 summarizes the major cellular and molecular events contributing to immune privilege in the eye.

### 2.1. Physical Immune Barriers

The BRB is of great significance for retinal immune privilege, in that it restricts the entry of circulating immune cells into the eye. It is composed of two parts: the inner and the outer BRB (Figure 4A). The outer BRB refers to the tight junctions between RPE cells, while retinal capillary endothelial cells comprise the inner BRB [36,37]. The outer BRB exerts immune privilege by secreting molecules, such as pigment epithelial-cell-derived factor and vascular endothelial growth factor (VEGF), to maintain the integrity of retinal vasculature and choriocapillaris [38,39]. The inner BRB includes pericytes, astrocytes, Müller cells, the collagen IV basement membrane, and perivascular macrophages [40,41] (Figure 4B). The second ocular barrier is the anterior barrier of the eye—the blood–aqueous barrier, which contains vascular endothelial cells in the iris and nonpigmented ciliary epithelium cells and prevents the translocation of substances from the plasma to the aqueous humor [42,43]. Breach of these physical barriers threatens immune privilege status and causes retinal disease, with increased BRB permeability and capillary hydrostatic pressure in the eye [44,45].

### 2.2. The Immune Regulatory System in the Eye

With the breakdown of physical barriers, the innate and adoptive immune responses in the intraocular microenvironment become activated and lead to immune suppression and inflammation inhibition, elicited by effector immune cells and endothelial cells. This immunological inhibition is achieved by soluble molecules from the aqueous humor and cell-membrane-bound molecules from other intraocular compartments. The soluble factors include transforming growth factor beta (TGF-β), alpha-melanocyte-stimulating hormone, calcitonin-gene-related peptides, thrombospondin-1, vasoactive intestinal peptides, and complement-inhibiting molecules [46].

Aqueous humor is a transparent, colorless fluid localized in the anterior and posterior chambers of the eye [47]. As one of the major ocular constituents, aqueous humor contains various immunomodulatory factors, including neuropeptides, cytokines, growth factors, and soluble cell-surface receptors to suppress inflammation responses and promote the induction of regulatory T cells [20,48,49]. Therefore, the roles of the innate and adoptive immune response are suppression of inflammation, upregulation of anti-inflammatory cytokines, and promotion of immune-tolerance-effector T cells and other immune effector cells [50,51,52,53,54].

Other crucial contributors to immune privilege are intraocular resident immune cells, which can inhibit immune responses and promote tolerance (as exemplified by macrophages and microglia). Antigens in the anterior chamber can systemically suppress immune response for a certain pathogen [55]. This phenomenon is also found in the vitreous cavity and subretinal space [56,57]. As an antigen-presenting cell with low expression levels of major histocompatibility complex (MHC) class II, the retinal resident microglia act as an immune checkpoint by preventing T-cell activation and induction of anti-inflammatory responses [46,58].

Ocular immune privilege is a double-edged sword. On the one hand, it makes the eye a microenvironment sealed from systemic circulation and protects all the optical components of the visual system; any minor breaches would cause devastating ocular diseases and failures of ocular allografts [24]. On the other hand, the rigidity of immune privilege, as exemplified by immunological tolerance of intraocular tumors and causes of infectious blindness (trachoma, river blindness, or herpes simplex virus), is a weakness [1,21].

## 3. Ocular Inflammation

Immune privilege is an evolutionary mechanism to protect system components, avoid damage from pathogens and injuries, and maintain ocular tissue integrity. However, once ocular immune privilege is breached, the entire ocular tissue is at high risk of damage from inflammation. Inflammation is a complex immune response of body tissues mediated by the innate immune system to combat deleterious insults and stimuli, including pathogens, physical injuries, irritants, and chemical compounds [59]. The purpose of inflammation is to eliminate the pathogen or the damaged cell and alert adjacent cells to elicit a broader immune response. A successful inflammatory response involves coordination between immune cells, endothelial cells, and neurons in the eye to elicit complex signal cascades through inflammatory factors. In the following section, we will summarize the roles of microglia, major inflammatory factors, and inflammatory pathways involved in ocular inflammation and ocular vascular diseases.

### 3.1. Microglia in the Retina

#### 3.1.1. Location and Morphology of Microglia in the Retina

Anatomically and developmentally, microglia originate from yolk-sac-derived myeloid progenitors, then they migrate into the central nervous system (CNS) prior to the formation of the blood–brain barrier. Microglia are specialized macrophages that reside in the CNS to maintain brain homeostasis through immune surveillance, phagocytosis, and the release of soluble factors [60]. The retina is an extension of the CNS; retinal microglia localize in the ganglion cell layer and plexiform layers in adult mouse retinas [61]. It has been shown that adult retinal microglia share a common developmental lineage but reside in two anatomically distinct locations, separated by interleukin (IL)-34 dependency and functional contribution to visual-information processing [62]. In the developing mouse, localization of retinal microglia coincides with the spatial distribution of the retinal synapses, starting at embryonic day 18, and become fixed to either the ganglion cell layer or one of the plexiform layers [63] at postnatal day 9 in mice. Dynamically, the density of retinal microglia also changes, with a significant decrease at birth followed by a rise during the first postnatal week and a decline again until postnatal day 28, before finally reaching a steady level in mice [64]. As the retina matures, retinal microglia morphology changes from amoeboid to ramified [65]. In developing human retinas, microglia density was the highest at 10-week-gestation age in the middle and deep layers of the peripheral retina. At 14-week-gestation age, high densities of microglia appear at the optic disc and microglia in the peripheral retina are redistributed to the more central retina [66]. Under pathophysiology conditions, in adult mice, full replenishment of retinal microglia takes approximately 6 months; this has been shown by studying bone marrow transplantation using enhanced green fluorescent protein-transgenic mice as donors for lethally irradiated C57BL/6 [67]. Additionally, a reduction in microglial numbers and function (tested using experimental microglial ablation in mice) can be restored by residual microglia in the central inner retina [68].

#### 3.1.2. Interaction between Microglia and Other Cells in the Retina

Physiologically, microglia stay in a homeostatic state with a ramified morphology, which is controlled by the interactions of microglia with other retinal cells (such as photoreceptors, RGCs, Müller cells, and endothelia cells) [69] through two well-known proteins:CD200 and fractalkine (also known as C-X3-C motif chemokine ligand 1 (CX3CL1)) [70,71,72,73,74]. In a mouse retina, fractalkine/CX3CR1 signaling plays an important role in the postnatal maturation of retinal photoreceptors as well as the rates of microglial processing and migration toward an injury [75,76]. The interaction between microglia and Müller cells is bidirectional; Müller cells can provide extracellular ATP for the microglial motility in healthy cells and release neurotrophic factors (such as the glial cell-line-derived neurotrophic factor, leukemia inhibitory factor, ciliary neurotrophic factor, nerve growth factor, neurotrophin 3, and basic fibroblast growth factor) that provide neuroprotection to photoreceptor cells under some circumstances [69,77,78]. The translocator protein is upregulated in retinal microglia under inflammatory conditions and interacts with its ligand, diazepam-binding inhibitor from astrocytes and Müller cells, by negatively regulating microglial activation [79,80].

In the presence of an insult, the microglia become reactive by changing their morphology, molecular expression profiles, and functions. Reactive microglia are highly migratory, proliferative, and phagocytic, releasing a variety of proinflammatory factors to trigger broader inflammatory responses, such as IL1, IL6, tumor necrosis factor alpha TNF-α, chemokines, and neurotrophic factors [81]. Phagocytic microglia can migrate toward the insult or damaged neurons and promote phagocytosis [82]. In general, short-term microglia activation is protective and results in the elimination of insults or damaged neurons in the CNS. Under chronic neuroinflammation, the BRB is breached, causing infiltration of monocyte-derived macrophages (MDMs) to the retina. It is reported that these MDMs shares some similarities in gene expression and morphology with retinal microglia, and are functionally responsive to local damage, though they are less responsive to retinal injuries and engaged in less migratory activity than microglia [83], as also reflected by transcriptional single-cell sorting and RNA sequencing in both animal models and human samples in retinal diseases [84,85] (Figure 5). Retinal microglia become reactive through the recognition of damage-associated molecular patterns or pathogen-associated molecular patterns present in the pathogens or damaged cells with corresponding receptors and pattern-recognition receptors. Receptors on the cell surface of microglia trigger signaling cascades for the clearance of damaged cell debris and pathogens to maintain retinal hemostasis [86], such as Toll-like receptors (TLRs), phagocytic receptors (complement receptor 3, CR3, and CR4), scavenger receptors (CD36 and CD91), and complement factors [87].

#### 3.1.3. Roles of Microglia in Ocular Vascular Diseases

As the eye contains immune privileged tissue, maintenance of retinal homeostasis and function is necessary for healthy vision. Any minor changes, due to disease or trauma, elicit a highly coordinated and cooperative response from the whole retinal cell community to minimize risk to the retinal neurons and restore a healthy state. However, when insults overwhelm the capacity of retinal hemostasis, retinal cell damage, dysfunction, and ultimately visual impairment result. This is manifested by chronic inflammation with activated retinal microglia, alternating levels of inflammatory factors, and dysregulation of signaling pathways. This section describes the pathological role of microglia in three ocular vascular diseases (ROP, AMD, and DR) associated with inflammation.

##### Retinopathy of Prematurity (ROP)

ROP is an alteration of immature retinal vascularization in preterm infants and one of the leading causes of vision loss in children globally. ROP is mainly manifested by inner retinal vasculopathy, but more studies have found that the disease is also related to choroidal degeneration, with defects in RPE and photoreceptor integrity as well as neuronal cell death [88,89,90,91]. Clinically, gestational age (<31 weeks of gestation) and birth weight (<1250 g) are the primary risk factors associated with ROP incidence [92,93]. Pathologically, ROP progress is divided into two phases: phase I involves delayed physiologic retinal vascular development, while phase II is manifested by increased vasoproliferation [94], which may subsequently lead to its most severe outcomes—retinal detachment and permanent vision loss. ROP has been extensively studied in humans and in several animal models, with the most widely used mouse models able to mimic the two distinct phases [95]. Established treatments (i.e., laser photocoagulation and anti-VEGF injection) continue to improve, while novel therapies are proposed (gene therapy and supplemental therapies) [96]. ROP is a multifactorial disease, as numerous factors (including oxygen, serum insulin-like growth factor-1, nutrition, and infections) can be involved in disease development [97,98]. Premature babies are more susceptible to infections because of their immature immune systems, and different stages of inflammation (prenatal, perinatal, and postnatal) are associated with ROP [99,100]. In a series of epidemiological studies, levels of proinflammatory cytokines changed in response to different infections in preterm infants [101,102], suggesting a relationship between ROP and inflammation.

Oxygen-induced retinopathy (OIR) is a well-established model for ROP [95]. Considerable evidence indicates that retinal microglia are involved in the OIR model, but their role in the OIR process is still unclear. It is reported that activation of retinal microglia correlates with retinal neovascularization in the OIR model [103]; reactive microglia were observed in the superficial layer of the central avascular zone from postnatal day (P) 8 to P12 and from P16 to P18, with a higher number seen around the vascular tufts, between the central avascular zone and the vascularized peripheral zone, and at P17 as resting microglia [103]. Intravitreal injection of bone-marrow-derived myeloid progenitor promotes retinal vascular repair in the OIR model, and these injected progenitor cells migrate to avascular regions of the retina and finally differentiate into microglia [104,105]. In the preclinical OIR mouse model, pharmacological inhibition of retinal microglia by clodronate (macrophage inhibitor) in both hyperoxic and hypoxic phases of OIR led to more severe retinal vascular damage [106]. Recently, a single-cell genomics study profiled highly heterogeneous microglia in an ROP mouse model, of which highly pathological microglia were spatially located within or around neovascular tufts [84]. Experimental data in animal models have not conclusively proven whether the microglia are beneficial or detrimental in ROP pathogenesis.

##### Age-Related Macular Degeneration (AMD)

AMD is an ocular disorder strongly associated with age, with incidence rates of 10% in people older than 65 years and >25% in people older than 75 years [107]. The disease is characterized by progressive loss of central vision due to retinal degeneration and neovascularization in the macula—a highly organized, oval-shaped pigmented area containing the highest density of cone photoreceptor cells. Dry AMD, also known as atrophic AMD, can be categorized as either early-, intermediate-, or late-stage depending on the appearance of the macular area [108]. Some intermediate-stage dry AMD patients have affected vison with detectable drusen changes and pigmentary disturbances. Drusen are produced by RPE during aging and localized at the interface between the RPE and Bruch’s membrane. In late-AMD (aka wet AMD or neovascular AMD) patients, central vision is severely impaired and there is neovascularization in the extracellular space (i.e., the subretinal space or the space between the RPE and Bruch’s membrane) [109,110]. Approximately 10–15% of AMD patients eventually advance from dry AMD to wet AMD. The contributory factors to the disease are complex, including age, family history/genetics, inflammation, oxidative stress, lifestyle, and diet [111,112,113,114]. There is increasing evidence that inflammation is associated with the pathogenesis of AMD [115,116,117]. This is further indicated by clinically detectable drusen in the subretinal space. The dysregulated immune system in the eyes constantly activates retinal microglia and inflammasome (proinflammatory factors and complement factors). It is reported that activated microglia contribute to macular degeneration by migrating to the outer nuclear layer and phagocytosing more photoreceptors cells in AMD patients [118].

Microglia have been implicated in early and intermediate AMD. Clinical samples from AMD patients indicate that reactive microglia accumulate on the apical surfaces of RPE cells, both overlying drusen and within drusen themselves [118,119], promoting drusenogenesis and actively contributing to the death of photoreceptors and RPE by secreting inflammatory factors [120,121]. In advanced AMD or neovascular AMD, microglial accumulation in the subretinal space is associated with vasculature growth of experimental CNV. In these animal models, subretinal-space-resident microglia secrete VEGF and other proangiogenic factors such as PDGF-β, FGF-1, FGF-2, and TGF-β1, promoting pathological choroidal blood vessel growth [121,122]. Microglia are required for the upregulation of inflammatory factors during retinal degeneration [123].

Macrophages present in both experimental and human CNV tissues. Wang et al. [124] report that bone-marrow-derived macrophages in bone marrow chimeric mice and local activated microglia quickly move into laser-induced CNV areas. *Ccr2^−/−^* mice exhibited smaller CMV size with a reduced number of ocular-infiltrating macrophages in an experimental CNV mouse model [125,126]. Macrophage subtypes are identified in *Ccr2^−/−^* mice using laser-induced CNV and the origin of macrophage subsets is mapped using *Cx3cr1^CreER^* mice; comparing to single-cell RNA-sequencing data of human choroid tissues with AMD, similar macrophage subtypes are found in both mice and human patients [127]. Single-cell RNA-sequencing data on immune cells from wild-type and *Ccr2^−/−^* mice with experimental CNV show that MDMs secrete VEGFA, IL1b, and other pathological cytokines that drive CNV, suggesting that macrophage subsets are critical drivers of CNV [128]. Therefore, targeting specific, macrophage subsets is a potential novel therapeutic for nAMD.

##### Diabetic Retinopathy (DR)

DR is a vision-threatening ocular disease affecting a primarily middle-aged working population with diabetes mellitus. It is the most common complication compared with other microvascular complications, such as diabetic nephropathy and neuropathy [129]. Globally, it is estimated that DR will affect 191 million people by 2030, and numbers continue to rise as the prevalence of diabetes increases [130,131]. DR is categorized as either early-stage nonproliferative DR or advanced-stage proliferative DR based on the presence of microvascular lesions. Specifically, nonproliferative DR is diagnosed by the presence of visible microaneurysms, retinal hemorrhages, intraretinal microvascular abnormalities, and venous caliber changes. By comparison, extensive pathological neovascularization develops in proliferative DR. The presence of diabetic macular edema across all DR severity levels makes it another ophthalmoscopically visible marker of DR. Diabetic macular edema is a fluid accumulation in the retina, produced from the breakdown of the BRB. The extravasation of fluid into the retina leads to abnormal retinal thickening and macula edema [132,133]. In the literature, inflammation and DR have been linked since 1967; it was found that diabetic patients taking salicylates to treat rheumatoid arthritis exhibit a low incidence of DR [134]. Since then, increasing evidence indicates a strong association between DR and inflammation [135,136,137]. In some risky conditions related to diabetes, such as hyperglycemia, hypertension, and obesity, microglia become activated [138,139,140]. In DR patients, microglia activate at different stages, increase in number, and migrate into the inner retinal layers with distributions noted around microaneurysms, intraretinal hemorrhages, and retinal and vitreal neovascularization [141]. In patients with diabetic macular edema, microglia infiltrate the outer retina and subretinal space, but the exact mechanism of microglia activation in the context of DR remains largely unknown. It is noted that MDMs also involve DR pathogenesis, as evidenced by the *Ccr2^−^^/−^* mice and *Ccl2^−^^/−^* mice with reduced DR pathogenesis [142,143], while *Cx3cr1^−^^/−^* mice have worsened DR pathology [144]. Overall, further investigations are needed into the role of microglia in the DR pathogenesis.

### 3.2. Inflammatory Mediators and Modulators

#### 3.2.1. Complement Systems

The complement system is an integral part of the innate immune system, playing an important role in the defense against pathogens and the maintenance of homeostasis in the eye. The complement system can be activated by at least three pathways: the classical pathway (CP), the mannose-binding lectin (MBL) pathway, and the alternative pathway (AP) [145]. Retinal microglia and RPE cells are the major cellular sources of local complement expression [146,147,148], but only select components of complement proteins and regulatory molecules are expressed, creating a sealed microenvironment independent of systemic complement [149]. Accumulating evidence suggests complement genes are upregulated in the retina, RPE, and choroid in normal aging mice [150,151,152,153] and in the choriocapillaris of healthy aged human eyes [154,155]. Over the past several decades, extensive investigations have found that single nucleotide polymorphisms (SNP) in the alternative complement pathway are involved in AMD, as summarized by Park et al. [156]. Under normal aging conditions, the complement system undergoes low-grade activation to maintain retinal homeostasis, although the physiological role of this activation is not yet fully understood. Complement system activation has been implicated in a wide range of ocular diseases [149,157].

Complement systems are critical in the pathology of vascular eye diseases. Several studies indicate that genetic variations in complement factor H (CFH) and complement components 2 (C2), C3, and C5 are associated with increased vulnerability to AMD [158,159,160,161,162,163,164,165,166]. Activated complement factors initiate an inflammatory cascade resulting in enhanced expression of proinflammatory molecules and an increase in opsonization and phagocytosis capability [167]. It has been shown that the levels of complement fragments C3a and Ba are upregulated in wet AMD patients [168], and soft drusen from AMD donors contain bioactive C3a and C5a, which both induce neovascularization [169]. Similarly, strong associations are found between CFH, CFB, and C3 variants and ROP occurrence in a clinical investigation, indicating a possible involvement of the alternative complement pathway in ROP [157]. In systemic investigation of DR patients, the level of C3, C3-activated fragment C3bα’, and CFH are upregulated in the vitreous of PDR patients [170], and a strong correlation exists between aqueous and vitreous complement levels in diabetic eye disease [171].

#### 3.2.2. Cytokines

Cytokines are small, secreted proteins released by cells to modulate intercellular communication. As the key modulators of inflammation, cytokines can be divided into several functional groups, including proinflammatory molecules, anti-inflammatory molecules, and cytokines involved in adaptive immunity [172,173]. As an immune-privileged tissue, the retina contains an abundance of cells, such as macrophages, microglia, and endothelial cells, which can be the source of cytokines. Recently, photoreceptors were reported to release cytokines [174]. Under pathological conditions, the level of cytokines changes accordingly to elicit a variety of signaling cascades for the upkeep of local retinal microenvironment homeostasis. The roles of various cytokines under different pathological conditions are summarized in Table 1. Notably, TGF-β and IL-10 are secreted from retinal cells under normal conditions to maintain the immune-privileged state of the eye through anti-inflammatory activities of these factors [175]. However, under pathological conditions, such as glaucoma, AMD, DR, and ROP, the levels of these cytokines change [168,176,177,178,179]. Müller cells are the major source of cytokines in response to infection and hyperglycemic conditions [180,181]. IL-33, secreted by Müller cells, is identified as a key regulator of inflammation and photoreceptor degeneration after retinal stress or injury [182]. The roles of cytokines in ocular vascular diseases are summarized in Table 1.

Increasing evidence indicates that cytokine profiles change in AMD patients, both in local ocular tissue and the systemic plasma [183,184,185,186,187,188,189], suggesting that systemic inflammation contributes to the pathology of AMD. In a cross-sectional study regarding CFH Y402H polymorphism (CC, CT, or TT variants) in AMD patients, the at-risk CC variant carriers had higher levels of IL-6, IL-18, and TNF-α [190]. Further study shows that the plasma IL-6 level significantly correlated with the geographic atrophy enlargement rate in AMD patients [191]. Mechanistically, evidence indicates that IL-6 promotes choroidal neovascularization [192], while long-term exposure to TNF-α and IL-18 causes the alternation of RPE morphology and differentiation and RPE cell swelling and subsequent atrophy, respectively [193,194].

**Table 1 ijms-24-12090-t001:** Pathological function of cytokines in retinopathies.

Factor Types	Genes	Diseases	Pathological Function	References
Chemokines	*CXCL1*	AMD, DR	Promotes neutrophil recruitment	[195,196,197]
	*CXCL10*	AMD	Antiangiogenic	[198]
	*CXCL12*	AMD, DR	Chemoattractant	[199]
	*CCL2*	AMD, DR	Knockout mice exhibit some features of AMD	[143,200,201,202]
	*CCL5*	AMD, DR	Promotes Th1 cell recruitment	[203]
	*CCL11*	AMD, DR	Proangiogenic	[204,205]
	*CCL24*	AMD	N/A	[206]
	*CX3CL1*	RP, AMD, DR,	Prolongs cone survival; involved in microglia activation and recruitment	[207,208,209]
Colony stimulating factors	*GCSF*	ROP, AMD, DR	Attenuates oxidative-stress-induced apoptosis; Neuroprotective; anti-inflammatory	[210,211,212]
	*GMCSF*	AMD, DR	N/A	[210,213]
	*MCSF*	ROP, AMD, DR	Proangiogenic	[214,215]
Interferons	*IFN-α*	AMD	Anti-inflammatory	[216]
	*IFN-β*	AMD	Immunoprotective	[217,218]
	*IFN-γ*	AMD	N/A	[219]
Interleukins	*IL-1α*	AMD, DR	Proinflammatory	[220,221]
	*IL-1β*	AMD, ROP, DR	Proinflammatory; induce rod degeneration in AMD model	[101,222,223]
	*IL-1Ra*	DR, ROP,	Antagonist for IL-1α, IL-1β	[157,221]
	*IL-2*	DR	ND	[224]
	*IL-4*	AMD, DR	Proangiogenic	[225,226]
	*IL-6*	ROP, AMD, DR	Proinflammatory; promote choroidal neovascularization	[102,227,228,229]
	*IL-8*	AMD, ROP, DR	Chemokine, proinflammatory	[168,228,230]
	*IL-10*	AMD, ROP, DR	Proangiogenic in AMD and ROP	[231,232]
	*IL-11*	DR	N/A	[233]
	*IL-12*	AMD, ROP, DR	Antiangiogenic	[183,234,235]
	*IL-13*	AMD, DR	Anti-inflammatory	[215,236,237]
	*IL-17A*	ROP, AMD, DR	Proinflammatory; proangiogenic	[238,239]
	*IL-18*	AMD, DR	N/A	[193,240]
	*IL-21*	AMD, DR	Promotes Th17-cell differentiation	[241,242]
	*IL-22*	AMD, DR	Anti-inflammatory	[242]
	*IL-23*	AMD, DR	Proinflammatory	[243,244]
	*IL-26*	DR	N/A	[245]
	*IL-27*	AMD, DR	Anti-inflammatory; antiangiogenic	[246,247]
	*IL-31*	DR	N/A	[248]
	*IL-33*	ROP, AMD	Pro-inflammatory in AMD; anti-inflammatory in RD	[249,250]
	*IL-35*	DR	Anti-inflammatory	[63,68]
	*IL-37*	ROP, DR,	Anti-inflammatory; proangiogenic	[251,252]
	*IL-38*	ROP	Antiangiogenic	[253]
Transforming growth factor	*TGF-β*	AMD, DR	Anti-inflammatory; antiangiogenic	[254,255]
Tumor necrosis factor	*TNF-α*	ROP, AMD, DR,	Alters RPE morphological changes and BRP breakdown; proinflammatory.	[256,257,258,259]

ROP, retinopathy of prematurity; AMD, age-related macular degeneration; DR, diabetic retinopathy. GMCSF, granulocyte-macrophage colony-stimulating factor; MCSF, macrophage colony stimulating factor; IFN, interferon; Th, T helper cells; N/A: not determined.

In clinical studies, levels of several aqueous proinflammatory cytokines (such as IL-1β, IL-2, IL-4, IL-5, IL-6, IL-8, IL-10, and interferon-γ) increase in DR patients [179,260,261,262]. Additionally, it was found that the levels of TNF-α and IL-6 were elevated in the plasma of DR patients, but this needs to be further validated in larger populations [263,264]. IL-1β and TNF-α participate in DR pathology through the initiation of retinal capillary cell death, while IL-6 can modulate RPE and microglial cells to attract microglial cells on RPE cells in streptozotocin-induced mice [257,265,266].

#### 3.2.3. Chemokines

In some ocular diseases, such as AMD and DR, chemokines are accumulated [143,267,268]. A transcriptome-wide study in the retinas of AMD patients revealed upregulated C-C motif chemokine ligand 2 (CCL2), CXCL1, CXCL10, and CXCL11 [269]. C-C motif chemokine receptor 3 (CCR3) is specifically expressed in choroidal neovascular endothelial cells in humans with AMD and is believed to be a therapeutic target for AMD [270,271,272]. A clinical study revealed that MCP-1/CCL2, CXCL10/interferon-γ-inducible protein 10 (IP-10), and stromal-cell-derived factor 1 (SDF-1) may participate in pathogenesis of both proliferative vitreoretinopathy and proliferative DR, as levels are elevated in DR patients [273], while in severe nonproliferative DR patients the levels of CCL5 and SDF-1α are significantly elevated [274].

#### 3.2.4. Cyclooxygenase (COX) Enzyme

The cyclooxygenase (COX) enzyme can catalyze the conversion of arachidonic acid into prostaglandins and plays a critical role in inflammation, cancer, and organ development [275]. It has been suggested that COX exists in two isoforms, COX-1 and COX-2, each having different regulatory mechanisms, cellular distributions, and functions. Compared with COX-1, COX-2 is an induced protein with multiple regulatory elements (such as NF-κB, Sp1, a TATA box, CAAT Enhancer Binding Protein Beta (C/EBP β), and cAMP response element-binding (CREB) consensus sequences) in its promoter region [276,277]. COX-1 predominantly expresses in microglia and participates in the neuroinflammatory process. COX-2 resides in hippocampal and cortical glutamatergic neurons [278]. In the retina, COX-2 expresses in RPE cells, ganglion cells, and retinal blood vessels, and its expression is induced by proinflammatory cytokines and NF-κB in RPE cells [279,280]. The COX-2 inhibitor, rofecoxib, can ameliorate pathological angiogenesis in oxygen-induced retinopathy mice.

It was shown that iNOS and COX-2 act together to contribute to retinal cell death in diabetes as well as to the development of diabetic retinopathy [281]. This is largely because both molecules are controlled by NF-κB activity during inflammation [282]. In addition, the product of iNOS, especially reactive NOS, can positively regulate COX2 expression and activity [282]. In streptozotocin-treated mice, it was found that expression of COX-2 elevates in the retinal cells, while inhibition of its activity blocks the secretion of prostaglandin E2, the product of the cyclooxygenase pathway, and the production of VEGF [283]. Additionally, iNOS and its product, NO, are increased in the retinas of diabetic animals, and upregulation of iNOS affects ganglion cell and both inner plexiform and outer nuclear layers [284,285,286]; suppression of iNOS reverses light-evoked vasodilation in diabetic retinas.

#### 3.2.5. Other Mediators

Second messengers are intracellular small molecules and ions that act as broadcasters of the extracellular signal, which begins with the interaction of a ligand and its receptor outside of the cell [287,288]. After the interaction, receptors undergo conformation changes and initiate a series of downstream enzymatic reactions to produce and release second messengers. Subsequently, second messengers diffuse to the target-cellular compartments and trigger responses. The fine-tuned regulation of second messengers is necessary for optimal cellular signal transduction and the dysregulation is associated with pathological conditions such as inflammation, autoimmunity, and cardiovascular disease [289,290,291]. One of the well-studied second messengers, 3′,5′-cyclic AMP (cAMP), has pathways that lead to both pro- and anti-inflammatory effects in different cell types [289]. One study showed that cAMP levels are critical in TNF-α-induced disruption of the blood–retinal barrier in primary bovine retinal endothelial cells [292]. Nitric oxide (NO) is a free gaseous molecule that is involved in neurotransmission, vasodilation, and immune regulation [293,294,295]. In the eye, NO maintains normal vision through the involvement of the phototransduction and acts as a vascular endothelial relaxant to control retinal blood flow [296]. It is synthesized by three isoforms of NO synthase (NOS): neuronal (nNOS, NOS I), inducible (iNOS, or NOSII), and endothelial (eNOS, or NOSIII) [297]. iNOS (NOS II) is mainly responsible for generating large amounts of NO using a paracrine signaling manner under pathological circumstances, including liver diseases, insulin resistance, obesity, and cardiovascular diseases [298]. iNOS is not constitutively expressed. In response to proinflammatory stimuli, iNOS is expressed in microglia, macrophages, and astrocytes [299,300]. Mechanistically, this enzyme and its concomitant NO production can mediate retinal apoptosis in ischemic proliferative retinopathy [301]. A study in the quail retina showed that amoeboid microglia expressed iNOS during normal development; meanwhile, lipopolysaccharide treatment increased iNOS expression in retinal explants [302].

### 3.3. Key Pathways during Retinal Inflammation

Inflammation is a complex immune response of body tissue mediated by the innate immune system to combat deleterious insults and stimuli, including pathogens, physical injuries and irritants, and chemical compounds [59]. This is a process that involves numerous immune cells, endothelial cells, and local tissue cells with complex cell–cell interactions mediated by cytokines, growth factors, eicosanoids, complement factors, and peptides [173]. A successful inflammation process can eliminate foreign pathogens or damaged cells and restore tissue homeostasis, but chronic inflammation can cause irreversible tissue damage and lead to diseases including cancer [303], diabetes [304], and autoimmune diseases [305]. In this section, we will summarize some pathways related to inflammation in the retina (Figure 6).

#### 3.3.1. Janus Kinase/Signal Transducers and Activators of Transcription (JAK/STAT3/SOCS3) Pathway

The JAK/STAT pathway plays an important role in both hematopoiesis and immune development [306,307,308]. As a primary signaling pathway response to cytokines and growth factors, dysregulation of this pathway is closely associated with cancer and inflammatory diseases [309,310]. Physiologically, the organisms develop several mechanisms to modulate JAK/STAT pathways, as constant activation is detrimental to heath. This is exemplified by suppressors of cytokine signaling (SOCS), which inhibits activated STATs and protein tyrosine phosphatases [311]. SOCS proteins inhibit the JAK/STAT pathway by interacting with STATs; for example, SOCS3 modulates STAT3 activation in response to cytokines through the gp130 receptor [312]. Specifically, gp130 forms the receptor complex with type I receptor in response to IL-6 family cytokines, which in turn phosphorylates and activates STAT3 through JAK pathways. Activated STAT3 then translocates into the nucleus and induces gene expression, including SOCS3 and IL-6. The resulting SOCS3 acts as a feedback inhibitor of the JAK/STAT pathway by binding to phosphorylated JAK and its receptor, causing the inhibition of STAT3 activation. In the retina, SOCS3 is expressed in the photoreceptor cells and plays an important role during photoreceptor cell differentiation through a temporal fine-tuning regulation of STAT3 activity [313,314]. RGC-specific SOCS3 deficiency promotes axonal regeneration of the injured optic nerve in adult mice via a gp130-dependent pathway [315]. The STAT3/SOCS3 pathway also plays a role in controlling the visual function through ubiquitin-dependent degradation of rhodopsin [314,316]. As infiltrating macrophages are a cardinal feature of inflammation, ablation of SOCS3 in myeloid cells shows that STAT3 is constantly activated in the murine model of AMD and DR [9,317] while inhibition of STAT3 activation can partially reverse the phenotype of choroidal neovascularization in mice; this suggests the contributory role of STAT3/SOCS3 in the development of disease. Vascular SOCS3 deletion using Tie2-Cre resulted in increased pathological retinal angiogenesis in the murine models of oxygen-induced retinopathy [318], indicating that SOCS3 may be acting as an endogenous antiangiogenic under pathological conditions. Subsequently, retinal neurons and glia deletion of SOCS3 promoted pathological retinal angiogenesis, mainly through the increased production of vascular endothelial growth factor (VEGF), which in turn exerted an angiogenic effect on endothelial cells [319]. In the mouse model of experimental autoimmune uveoretinitis, myeloid-specific SOCS3 deletion promoted retinal degeneration and angiogenesis through arginase-1 [320]. Therefore, the STAT3/SOCS3 axis may act as a mediator between inflammation and angiogenesis under pathological conditions.

#### 3.3.2. c-Jun N-Terminal Kinases/Activator Protein 1 (AP-1) Pathway

The c-Jun N-terminal kinases (JNKs)/activator protein 1 (AP-1) pathway involves many physiological and pathological cellular activities, including gene expression, cell proliferation and differentiation, migration, apoptosis to inflammation, and carcinogenesis [321,322,323,324]. JNKs are also known as stress-activated protein kinases. A variety of stimuli—including inflammatory cytokines, growth factors, radiation, pathogens, heat shock, and genotoxic agents—will activate the JNK signaling pathway through the MAPK activation process. The phosphorylated JNKs then regulate the activities of AP-1 as well as the expression patterns of some apoptosis-related genes and microtubule-associated genes. [325]. Of note, the AP-1 transcription factor family contains c-Jun, c-Fos, activating transcription factor, and musculoaponeurotic fibrosarcoma protein families and functions through heterodimers or homodimers formed between different combinations with the AP-1 family [326]. In the retina, JNK signaling pathways are of great significance in both physiological and pathological conditions. JNK1 involves retinogenesis by increasing photoreceptor numbers in *Jnk1* knockout mice [327]. Under pathological conditions, genetic and pharmacological inhibition of JNK1 can ameliorate oxygen-induced retinopathy (a mouse model of ROP) by regulating the VEGF expression [328]. In an AMD model, JNK1 ablation alleviates the progression of disease manifested by decreased inflammation, reduced choroidal neovascularization, lower levels of choroidal VEGF, and a substantial reduction in choroidal apoptosis [329]. In a retinal angiogenesis model (very low-density lipoprotein receptor knockout (*Vldlr^−/−^*) mice), immune privilege in the photoreceptor layer and subretinal space are broken down, and abnormal blood vessels grow into the photoreceptor layer—normally an avascular region. In this model, the c-Fos level is induced in the photoreceptor layer, and increased c-Fos promotes angiogenesis—activating the STAT3/VEGFA pathway by producing cytokines of IL-6 and TNF-α [174].

Photoreceptors comprise rod and cone cells and they are also involved in retinal inflammation through the secretion of proinflammatory proteins, including iLs (1α, 1β, 6, and 12), CXCLs (1 and 12a), CCL25, MCP-1/CCL2, and TNF-α in the mouse model of diabetes, resulting in retinal vascular permeability [330,331]. It was reported that *Opsin* deletion in mice, which genetically mimics photoreceptor degeneration, prevents the diabetes-induced production of proinflammatory enzymes (such as iNOS and intercellular adhesion molecule 1 (ICAM-1) in the retina) [332], suggesting that inflammation originating from photoreceptors is of great importance in the pathogenesis of diabetic retinopathy. This was confirmed by an in vitro study using 661W photoreceptor cells, which showed that soluble factors released from photoreceptors in elevated glucose can stimulate TNF-α in leukocytes and endothelial cells [333]. Additionally, Sun et al. found the inflammatory regulator, c-Fos, was upregulated in *Vldlr^−/−^* mice, in which immune privilege was broken down with pathological retinal blood vessels invading photoreceptors. Further experiments indicated that photoreceptor c-Fos regulates retinal angiogenesis in retinal outer nuclear layers via the STAT3/VEGFA pathway [174]. This demonstrated the roles of photoreceptors in retinal inflammation under pathological conditions—as in the model of autosomal dominant retinitis pigmentosa, rods secrete TNF-α and signal a “self-destructive” program to the cones, resulting in their cell death [334].

#### 3.3.3. Nuclear Factor Kappa B (NF-κB) Pathway

The Toll-like receptor (TLR) pathway is developed evolutionarily in an organism to both sense a variety of pathogens and trigger immune responses. The TLRs recognize pathogen-associated molecular patterns and damage-associated molecular patterns present in various pathogens and damaged cells, respectively. This recognition relies on the leucine-rich repeat domain of TLRs by first initiating the recruitment of Toll-interleulin-1-receptor (TIR)-domain-containing adaptors [335]; subsequently activating and culminating NF-κB for the induction of inflammatory cytokines, chemokines, and type I interferons (IFNs); and ultimately protecting the host from the invading pathogens and eliminating the damaged cells. NF-κB is a ubiquitous transcription factor that is involved in a variety of biological activities and plays important roles in the immune regulation by modulating the expression of proinflammatory cytokines, chemokines, immunoreceptors, and enzymes [336,337]. In the avian retina, NF-κB acts as a hub, coordinating signals from microglia to regulate the formation of proliferating Müller glia-derived progenitor cells [338]. In some genetically or physically induced mouse models, NF-κB is constantly activated [339,340,341,342]. One study showed that the expression levels of phosphorylated IκBα and phosphorylated p65 increased in OIR mice, indicating constant activation of the NF-κB signaling pathways in this pathological condition. Subsequent NF-κB signaling inhibitor treatment reverses retinal neovascularization [343]. In addition, NF-κB p65 activation is found in retinal pericytes of diabetic donors; such activation can be associated with retinal microvascular cell apoptosis and capillary cellularity [344].

Using human retinal endothelial cells, one study found that hyperglycemia or euglycemia treatment induces TLR2 and TLR4 expression and subsequent NF-κB p65 activation—thus, producing increased levels of IL-8, IL-1β, TNF-α, MCP-1/CCL2, and enhanced monocyte adhesion to human retinal endothelial cells—while TLR-4/2 inhibition attenuates these phenotypes [345].

#### 3.3.4. VEGF Pathway

Abnormal pathological angiogenesis is the cardinal feature shared among vascular ocular diseases, including age-related macular degeneration, diabetic retinopathy and/or diabetic macular edema, retinal vein occlusion with macular edema as manifested by neovascularization, vascular permeability, and inflammation. Elevated levels of VEGF are secreted by retinal pigmented epithelium (RPE) [346], astrocytes [347], Müller cells [348], vascular endothelium [349], and ganglion cells [350] in response to hypoxic conditions, and macrophages in CNV models [351], resulting in the binding of VEGF to its receptor, VEGFR1/2. Upon binding, autophosphorylation of VEGFR2 is elicited, migration and proliferation of vascular endothelial cells are induced, pericytes are reorganized and detached from the vessel wall, and then vascular permeability is increased [352,353]. Based on these reports, anti-VEGF therapy emerged. The first anti-VEGF therapy was approved for clinical use in 2004 [354,355]. Over the years, several anti-VEGF agents have been introduced to clinics to treat vascular eye diseases [356,357], including ranibizumab, aflibercept, brolucizumab, and bevacizumab.

#### 3.3.5. Angiopoietin–Tie (Ang/Tie) Pathway

Ang/Tie signaling is another pathway important for vascular physiology, both in retinal homeostasis and pathology. Major components of this pathway include Ig (immunoglobulin) and EGF (epidermal growth factor) homology domains 2 (Tie2) receptor 1 and 2, angiopoietin-1 (Ang1) and angiopoietin-2 (Ang2), and the vascular endothelial tyrosine phosphatase (VE-PTP) receptor. Ang1 is considered an agonistic ligand. Binding of Ang1 to Tie2 induces autophosphorylation of the Tie2 receptor on endothelial cells and subsequently activates downstream pathways, Akt (or protein kinase B) and phosphatidylinositol 3-kinase (PI3K) pathways [358,359], which are responsible for endothelial cell survival. Meanwhile, Akt activation suppresses forkhead transcription factor FKHR (FOXO1) activity and leads to vascular destabilization and remodeling by regulating Ang2 [360]. Moreover, Tie2 activation elicits the anti-inflammatory effect through the A20 binding inhibitor of NF-κB activation-2, ABIN-2, an inhibitor of NF-κB–mediated inflammatory gene expression [361]. As an antagonist, the binding of Ang2 can block Tie2 phosphorylation induced by Ang1, leading to abnormal vascular structure formation, increased vascular permeability, and increased inflammation [353,362,363]. Accumulating evidence support the fact that Ang2 has a context-dependent role, as it serves as a Tie2 receptor antagonist in resting ECs [353,364], but under pathological conditions it can act as a compensatory factor to increase Tie2 activity and angiogenesis [365]. Numerous reports—about levels of Ang1 and Ang2 increasing in diabetic retinopathy [366,367], AMD [368], ROP [369], and upregulation of vascular endothelial-protein tyrosine phosphatase (VE-PTP) in retinal neovascularization [370]—indicate that this pathway is a promising compensational therapeutical target for vascular ocular diseases, in concert with anti-VEGF therapy [371]. Several biological molecules, including AXT107, faricimab, razuprotafib, and nesvacumab, have been developed and evaluated [353,372], and faricimab, a bispecific antibody targeting VEGF and Ang2, completed a phase-3 clinical trial and has been FDA-approved for the treatment of wet AMD and diabetic macular edema in 2022 [373,374].

## 4. Conclusions

This review summarizes recent clinical and research advances on inflammation in retina. As one of the most metabolically dynamic and versatile tissues mainly composed of neurons and glia, the eye develops complicated mechanisms to preserve the retina neurons and their circuits in an immune-privileged state against a variety of internal and external insults. Some key inflammatory effectors, microglia and inflammatory pathways, are induced and activated if immune privilege breaks down, thereby eliminating insults quickly and avoiding retinal impairment. However, some ocular diseases (such as retinopathies) provoke chronic inflammation, which puts all retinal cells under inflammatory stress—as proven by constantly activated microglia and altered levels of inflammatory factors and pathways. These factors are a great danger to the retina as they can lead to photoreceptor cell death, pathological neovascularization, vascular leakage, and consequently vision impairment.

There is now convincing evidence that chronic inflammation is a hallmark of ocular retinal diseases. Extensive and broad investigations combining clinical research, in vitro experiments, and animal models have contributed to our understanding of the more detailed mechanisms of inflammation under various pathological conditions. The accumulation of experimental proofs in animal models indicate that pharmacological interference with some effectors (such as microglia, VEGF/Ang, complement pathway inhibitors, and visual cycle inhibitors) can ameliorate or delay retinal disease [375]. A variety of compounds targeting microglia, either by depleting microglia, reprograming activated microglia to a less harmful state, or blockading downstream events caused by microglia activation, have been implicated in the preclinical animal models in AMD and DR diseases [69]. Although microglia act as a health guard to physiologically protect the retina from invasion, recent research indicates that microglia may perform a slightly different function, based on different stages of disease progression and physical localization of the microglia population in the retina [376]. Therefore, more research is needed on the detailed mechanisms of the microglial–endothelial cell and microglial–retinal neuron interactions to further develop targeted pharmaceuticals. Another promising alternative approach is stem cell therapy, in which transplantation of human pluripotent stem cells (hPSC)-derived RPE cells can improve vision in AMD patients [377]; but, to date there are very limited data that must be validated in clinical trials.

Immune privilege and inflammation act as two indispensable sides of the same coin to safeguard retinal homeostasis. There is still much to understand about the complex and delicate role of inflammation in retinal disease pathology. This may best be accomplished using both animal models and human retinal tissues (especially given the increasing accessibility of human tissue banks), as minor differences between the two approaches can impede translational studies. Thorough exploration of inflammation will bring new breakthroughs for the treatment of retinal disease in the future.

## Figures and Tables

**Figure 1 ijms-24-12090-f001:**
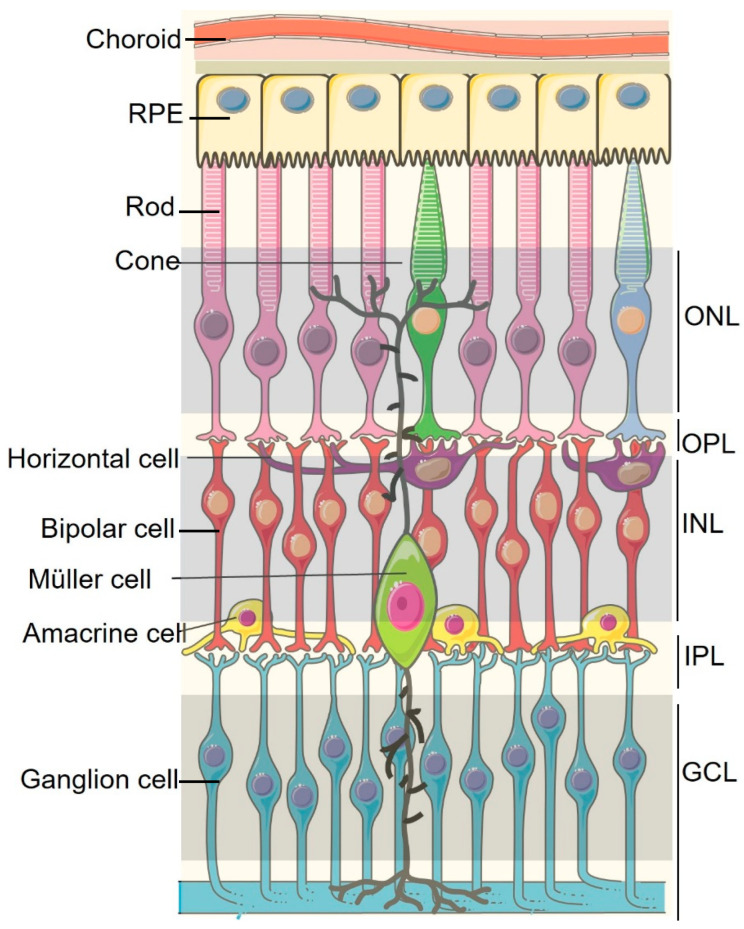
Schematic illustration of the cell types in the retina and their retinal layers. The neural retina cell types include RPE cells, photoreceptors (rods and cones), horizontal cells, bipolar cells, amacrine cells, ganglion cells and supporting cells, Müller cells, RPE, and retinal pigment epithelium. ONL, outer nuclear layer; OPL, outer plexiform layer; INL, inner nuclear layer; IPL, inner plexiform layer; GCL, ganglion cell layer. The figure was partly generated using Servier Medical Art, provided by Servier, licensed under a Creative Commons Attribution 3.0.

**Figure 2 ijms-24-12090-f002:**
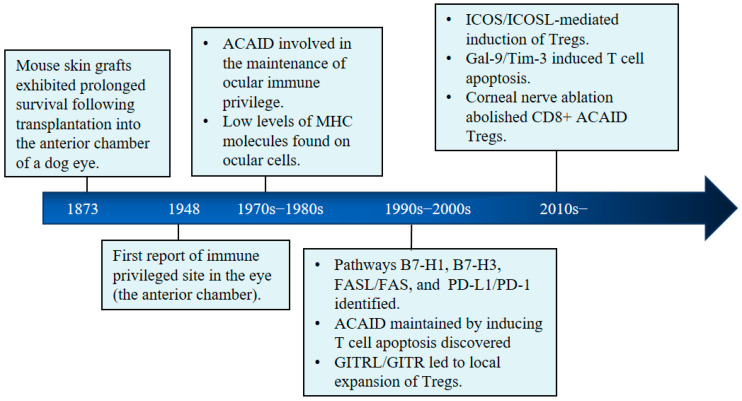
History of research on immune privilege in the eye [12,26,27,28,29,30,31,32,33,34,35]. ACAID, anterior chamber-associated immune deviation; MHC, major histocompatibility complex; ICOS, inducible costimulatory molecule; ICOSL, ICOS ligand (also known as B7-related protein (B7RP-1)); Tregs, regulatory T cells; Gal-9/Tim-3, Galectin-9/T-cell immunoglobulin and mucin domain-3; CD8, T cell subset, B7-H1, B7 homolog 1; B7-H3, B7 homology 3; FASL, FAS ligand; FAS, FS-7-associated surface antigen; PD-L1, programmed death ligand 1; PD-1, programmed cell death protein 1; GITRL, GITR ligand; GITR, glucocorticoid-induced TNF-receptor-family-related protein.

**Figure 3 ijms-24-12090-f003:**
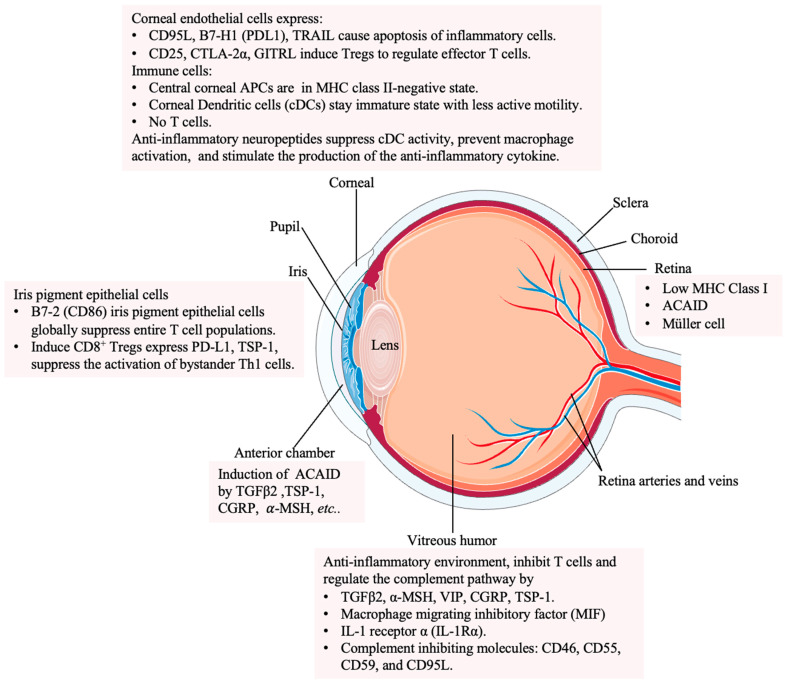
Contributing molecular mechanisms of ocular immune privilege. In addition to physical barriers, a variety of endothelial cells, immune cells, and retinal neuronal cells provide the second layer of protection by releasing immunosuppressive factors; antigenic proteins cause anterior-chamber-associated immune deviation, acting as a third layer of protection. Antigen-presenting cells (APCs); tumor-necrosis-factor-related apoptosis-inducing ligand (TRAIL); cytotoxic-T-lymphocyte-associated antigen-2 alpha (CTLA-2α: a cystein proteinase inhibitor); glucocorticoid-induced tumor necrosis factor (TNF)-receptor-family-related protein ligand (GITRL); thrombospondin-1 (TSP-1); VIP (vasoactive intestinal peptide); CGRP (calcitonin gene related peptide). (The figure was partly generated using Servier Medical Art, provided by Servier, licensed under a Creative Commons Attribution 3.0 unported license).

**Figure 4 ijms-24-12090-f004:**
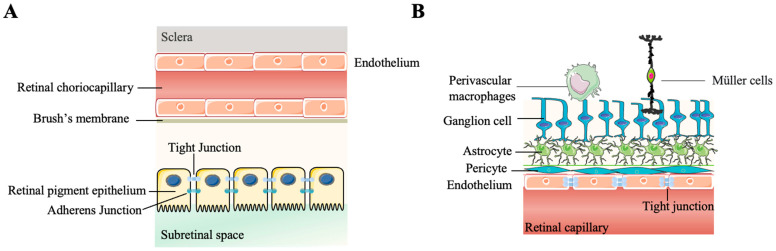
Schematic diagram of the blood–retinal barrier (BRB). The BRB is composed of outer BRB (**A**) and inner BRB (**B**), which serve as physical barriers in immune privilege to isolate the ocular compartment from circulation. (This figure was partly generated using Servier Medical Art, provided by Servier, licensed under a Creative Commons Attribution 3.0 unported license).

**Figure 5 ijms-24-12090-f005:**
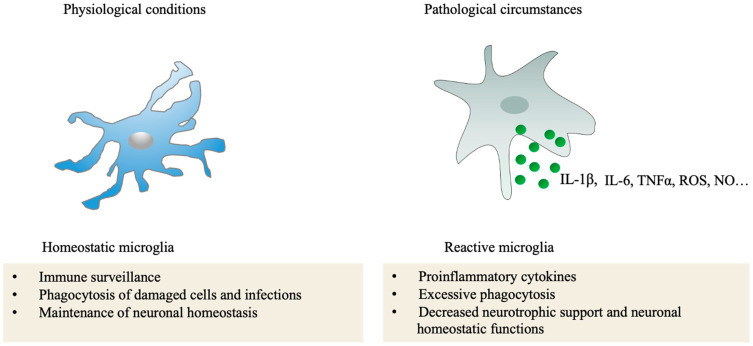
Functions of microglia in healthy retinas and pathological circumstances. In normal conditions, microglia play an important roles in tissue homeostasis. In pathological conditions, microglia abnormal activation triggers more inflammation and excessive phagocytosis. (This figure was partly generated using Servier Medical Art, provided by Servier, licensed under a Creative Commons Attribution 3.0 unported license).

**Figure 6 ijms-24-12090-f006:**
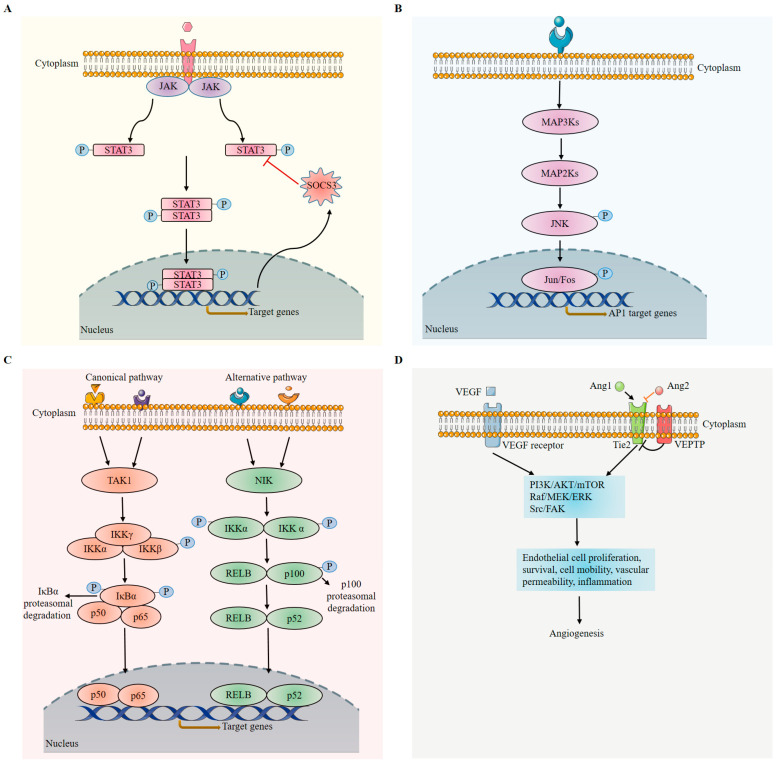
Major pathways involved in retinal inflammation. (**A**) JAK/STAT3/SOCS3 pathway. Cytokines (such as IL-6) bind to their cognate receptors, the receptor-associated tyrosine kinases (Janus kinases) lead to phosphorylation and dimerization of signal transducers and activators of transcription (STATs), which then translocate to the nucleus to induce expression of proinflammatory genes. As the target gene of this pathway, SOCS3 acts as a negative feedback loop to inhibit JAKs and block the signaling pathway. (**B**) Growth factors, cytokines, and pathogens induce activation of members of the mitogen-activated protein kinase kinase kinase (MAP3K) family, followed by sequential phosphorylation of MAP2K and JNK. Activated JNK phosphorylates Jun/Fos result in the expression of target genes. (**C**). Canonical and alternative NF-κB pathways. Toll like receptors (TLRs) and proinflammatory cytokines (such as TNF-α and IL-1) activate canonical pathway. Upon stimulation, TGFβ-activated kinase 1 (TAK1) phosphorylates IKKβ in the inhibitor of κB kinase (IKK) complex, followed by phosphorylation of IκBα, which sequester NF-κB members in the cytoplasm. The phosphorylated IκBα is degraded by the proteasome, leading to the translocation of the NF-κB dimers p65/p50 complex to the nucleus where it activates the transcription of target genes. In noncanonical pathway, NF-kB-inducing kinase (NIK) activates IKKα after stimulation, followed by the phosphorylation of p100. The phosphorylated p100 is processed in proteasome, generating the subunit p52. Finally, the p52/RelB NF-κB complex is able to translocate to the nucleus and induce gene expression. (**D**) In endothelial cells, binding of VEGF to its receptor activates PI3K/AKT/mTOR, Raf/MERK/ERK, and src/FAK pathways, leading to cell proliferation, survival, vascular permeability, and inflammation, therefore, promoting angiogenesis. Angiopoietin (Ang) binds its receptor Tie. Ang1 binds to Tie2 to activate the signaling pathway, while the Ang2 binding or Tie2/VE-PTP interaction blocks pathway. (This figure was partly generated using Servier Medical Art, provided by Servier, licensed under a Creative Commons Attribution 3.0 unported license).

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
