# Peer review of "Ocular Vascular Diseases: From Retinal Immune Privilege to Inflammation"

_ijms, 2023, doi:10.3390/ijms241512090_

Round 1
Reviewer 1 Report
The review article titled "Ocular vascular diseases: From retinal immune privilege to inflammation" by Wang et al. provides a comprehensive summary of the immune privilege of the eye and its implications in ocular diseases. The authors discuss the consequences of breaching immune privilege, emphasizing the activation of local microglia and release of immunological factors. While acute inflammatory responses are vital for defense, chronic inflammation can lead to retinal tissue damage, dysfunction, and vision loss. The review also covers recent advances in ocular immunity and vascular diseases, making it a valuable resource. I believe this manuscript offers a valuable examination of the field.
Line 40: Muller cells and ????. Some word is missing here.
It is advisable to make revisions to Figure 3 to ensure the accurate alignment of labels with the three main mechanisms. The current version of Figure 3 does not adequately represent the three mechanisms in a clear and sequential order, emphasizing the need for improvements in the visual presentation.
To enhance the comprehension and visual representation of the extensive discussion on the important function of microglia in the retina, it is highly recommended to include a dedicated figure in paragraph 3.1, as the absence of visual representations such as tables or figures in this paragraph limits the clarity and impact of the information presented.
Line 181 and 193. Currently it is all about mouse. It would be valuable to include information about human retinal microglia to extend the discussion and expand our understanding of this topic.
Line 213: Typo in this line.
Line 441. It should be 3.3.4, not 3.4
Line 442: It should be 3.3.4.1
For section 3.3, despite the public availability of numerous figures illustrating these pathways individually, it would be highly valuable to present side-by-side figure panels depicting these pathways in the ocular system, allowing for a comprehensive visual presentation and facilitating a better understanding of their interconnections and implications within the context of ocular physiology.
Minor revision needed. Overall, a very quality review article with sufficient information provided.
Author Response
See response letter in attached file.

Reviewer 2 Report
In my opinion the review is well written and I do not think it needs some more extra work.
The topic, inflammation in the retina, is clearly exposed and report all the main resent clinical and research advances. My colleague (reviewer n.2) suggest to add a dedicated paragraph related to the microglia and monocyte-derived macrophages (MDMs), this addition will definitely increase the quality of the review.
Overall from my point of view is a good review, also the table and the images provided are useful for the reader.
Author Response
See response letter in attached file.

Reviewer 3 Report
Wang et al present a well-written review on ocular vascular disease, immune privilege, and inflammation. My main concern with this review is the lack of discussion and discrimination between microglia and monocyte-derived macrophages (MDMs). I would recommend either a section devoted to MDMs or explicit discussion of microglia vs MDM in each disease context, as discussed below. In several sections, new single-cell RNA-seq experiments have been omitted and are important to be included.
1. Line 217: “However, constant activation of microglia, as in chronic neuroinflammation, puts all CNS neurons in a high risk environment, causing undesirable neuronal cell death and triggering the onset of diseases (73, 74).” This is not so straightforward because during chronic neuroinflammation, the BRB is breached, causing MDMs to infiltrate the retina. Once MDMs infiltrate the retina, their expression becomes more microglia-like over time (PMID: 36510226) and it is difficult to determine if the MDMs or reactive microglia lead to damage. This nuance should be stated. This section also should mention disease-associated microglia (PMID: 28602351) and their transcriptomic phenotype.
2. The role of microglia in AMD are not so straightforward, many earlier studies have difficulty distinguishing microglia from MDMs. The authors should discuss studies using Ccr2-/- mice which show smaller CNV sizes (PMID: 12832439, 27532664, 33187533), implying a critical component of MDMs to the laser-induced CNV model. Microglia have been found in CNV lesions but no genetic studies have targeted them. The only genetic study in Cx3cr1-/- mice, which have dysregulated microglia, shows larger lesions, perhaps suggesting that microglia inhibit CNV, but this is not definitive (PMID: 17909628). Recently scRNA-seq studies show that pathological macrophages in the laser-induced CNV model are MDMs in agreement with Ccr2-/- studies. These MDMs secrete VEGFA, IL1b and other pathological cytokines that drive CNV. Microglia in this scRNA-seq were not enriched for angiogenic GO terms (36821388), suggesting that MDMs are critical drivers of CNV over microglia in agreement with genetic studies.
3. The ROP section should discuss new scRNA-seq study where a subset of pathological microglia were identified (PMID: 36264636).
4. The microglia in DR section needs to also highlight the roles of classical MDMs and microglia in DR. It should be noted that both Ccr2-/- and Ccl2-/- mice have reduced DR pathogenesis, suggesting that MDMs are associated with DR worsening (PMID: 36698021, 25329075). It should also be noted that Cx3cr1-/- mice have worsened DR and microglia are abnormal in this model (PMID 27344677), possibly suggesting that microglia inhibit DR pathogenesis.
5. Line 188: “In adult mice, full replenishment of retinal microglia takes approximately 6 months; this has been shown by studying bone marrow transplantation using enhanced green fluorescent protein-transgenic mice as donors for lethally irradiated C57BL/6.” This sentence implies physiology but this is not physiology but pathophysiology. Retinal microglia self-renew and are not replenished during physiologic conditions, and this should be explicitly stated to not confuse the reader.
6. Second messengers and NFkB: please do not discuss the outdated M1-M2 nomenclature that is not replicable in vivo in any tissue known to date using scRNA-seq. In the eye, multiple scRNA-seq studies have been done, which show multiple macrophage transcriptional phenotypes but there are always more than 2 and they do not fit into the M1-M2 dichotomy. The following sentence in the second messengers section in fact highlights the discrepancy. If cAMP is important for M1->M2 polarization, then why does cAMP increase TNFa, a canonical M1 cytokine? I would recommend simply stating what the data shows rather than forcing the data into the outdated and artificial M1-M2 dichotomy.
7. The inner BRB includes more than just ECs. It includes pericytes, astrocytes, muller glia, the collagen IV basement membrane, and perivascular macrophages (PMID: 19608545, 35941655), which I think should be specifically mentioned given the focus of this review on inflammation.
8. Chemokines: MCP1 and CCL2 are in fact the same chemokine linked to DR and AMD. Please choose one nomenclature rather than interchanging them.
9. TNF-alpha is routinely TNF@ or other strange symbols in this manuscript
10. VEGF: also expressed by macrophages (PMID: 24714223).
Good, minor typos only
Author Response
See response letter in attached file.
